# Using Diatom and Apicomplexan Models to Study the Heme Pathway of *Chromera velia*

**DOI:** 10.3390/ijms22126495

**Published:** 2021-06-17

**Authors:** Jitka Richtová, Lilach Sheiner, Ansgar Gruber, Shun-Min Yang, Luděk Kořený, Boris Striepen, Miroslav Oborník

**Affiliations:** 1Biology Centre CAS, Laboratory of Evolutionary Protistology, Institute of Parasitology, 370 05 České Budějovice, Czech Republic; jitka.richtova@paru.cas.cz (J.R.); ansgar.gruber@paru.cas.cz (A.G.); shun-min.yang@paru.cas.cz (S.-M.Y.); 2Faculty of Science, University of South Bohemia, 370 05 České Budějovice, Czech Republic; 3Welcome Centre for Integrative Parasitology, College of Medical, Veterinary and Life Sciences, Institute of Infection, Immunity and Inflammation, University of Glasgow, Glasgow G12 8QQ, UK; lilach.sheiner@glasgow.ac.uk; 4Department of Biochemistry, University of Cambridge, Cambridge CB2 1TN, UK; lk360@cam.ac.uk; 5Department of Pathobiology, School of Veterinary Medicine, University of Pennsylvania, Philadelphia, PA 19104, USA; striepen@vet.upenn.edu

**Keywords:** tetrapyrrole biosynthesis, heterologous expression, *Chromera velia*, predictions

## Abstract

Heme biosynthesis is essential for almost all living organisms. Despite its conserved function, the pathway’s enzymes can be located in a remarkable diversity of cellular compartments in different organisms. This location does not always reflect their evolutionary origins, as might be expected from the history of their acquisition through endosymbiosis. Instead, the final subcellular localization of the enzyme reflects multiple factors, including evolutionary origin, demand for the product, availability of the substrate, and mechanism of pathway regulation. The biosynthesis of heme in the apicomonad *Chromera velia* follows a chimeric pathway combining heme elements from the ancient algal symbiont and the host. Computational analyses using different algorithms predict complex targeting patterns, placing enzymes in the mitochondrion, plastid, endoplasmic reticulum, or the cytoplasm. We employed heterologous reporter gene expression in the apicomplexan parasite *Toxoplasma gondii* and the diatom *Phaeodactylum tricornutum* to experimentally test these predictions. 5-aminolevulinate synthase was located in the mitochondria in both transfection systems. In *T. gondii*, the two 5-aminolevulinate dehydratases were located in the cytosol, uroporphyrinogen synthase in the mitochondrion, and the two ferrochelatases in the plastid. In *P. tricornutum*, all remaining enzymes, from ALA-dehydratase to ferrochelatase, were placed either in the endoplasmic reticulum or in the periplastidial space.

## 1. Introduction

Life as we know it, would not be possible without tetrapyrroles, namely chlorophyll and heme. While chlorophyll is used exclusively in photosynthesis, heme can be involved in various electron transport chains and redox reactions [1]. Heme appears to be essential for almost all life on Earth, with only a few exceptions among pathogenic and anaerobic bacteria, and a single exception in aerobic eukaryotes, the kinetoplastid *Phytomonas serpens* [2]. All other organisms either synthesize their own heme or obtain it from external sources [2]. Both heme and chlorophyll share a common synthetic pathway (up to protoporphyrinogen IX), which is well conserved among all three domains of life [3] (outlined in Figure 1). The first precursor of this pathway, 5-aminolevulinic acid (ALA), can be synthesized in two fundamentally different ways: primary heterotrophic eukaryotes and Alphaproteobacteria use the C4 (or Shemin) pathway, the condensation of succinyl-CoA and glycine, while Eubacteria, Archaea, and eukaryotic phototrophs form ALA from glutamate via a set of reactions termed the C5 pathway [4]. Eight molecules of ALA are assembled in three consecutive steps to uroporphyrinogen III, the first macrocyclic tetrapyrrole, which can convert to siroheme, or, alternatively, the next three steps of the synthesis lead to protoporphyrinogen IX. In the chlorophyll synthesis branch, magnesium-chelatase inserts an Mg^2+^ ion into the center of the porphyrin ring. In the heme synthesis branch, insertion of a Fe^2+^ ion into the ring by ferrochelatase (FECH) finally completes the heme [1].

Tetrapyrrole biosynthesis in eukaryotes is largely influenced by past endosymbiotic events, in which mitochondria and plastids were acquired. This is reflected in the phylogenetic affinities of the associated genes, that often demonstrate similarity to homologous genes in Alphaproteobacteria or cyanobacteria, for mitochondrial or a plastid origin, respectively [4,5]. While the tetrapyrrole pathway is almost universally present, the subcellular distribution of the enzymes differs widely across the eukaryotic biodiversity. Location corresponds to the trophic strategy of the organism, cellular demand for the final products of the pathway, the evolutionary origin of the enzyme, and the need for tight regulation of the pathway [6,7,8,9,10].

In primary eukaryotic heterotrophs, both the initial and terminal steps of the synthesis take place in the mitochondria, which is not surprising considering the availability of the precursor, succinyl-CoA, and the demand for heme in the cytochromes of the respiratory chain [4,5,11]. The common location for the start and completion of heme synthesis is also important for the regulation of the pathway, which is mainly achieved by the heme-mediated inhibition of ALA formation [6,7,8,9,10]. The middle part of the pathway in heterotrophs takes place in the cytosol, which necessitates the transport of ALA and a porphyrin intermediate across the mitochondrial membranes [12,13]. Most phototrophs use the C5 pathway to begin the tetrapyrrole synthesis, and the whole process is located inside the plastid, the place with the highest demand for the final products, chlorophyll, and heme [14]. The euglenid alga *Euglena gracilis* [15] and the chlorarachniophyte *Bigelowiella natans* [16] possess both the plastid located (C5 based) pathway, and the mitochondrially-cytosolic (C4 based) pathway. Apicomplexan parasites [17] such as *Plasmodium* or *Toxoplasma* harbor a non-photosynthetic relic plastid (the apicoplast) and possess a rather peculiar heme synthesis. The pathway starts via the C4 route in the mitochondrion, the next four steps are apicoplast localized, consecutively, coproporphyrinogen oxidase (CPOX) is active in the cytosol, and the synthesis is completed by protoporphyrinogen oxidase (PPOX) and FECH in the mitochondrion again [5,11,18,19,20,21]. Such complicated intracellular distribution of heme pathway enzymes most likely arose because of the transition from a photosynthetic to a parasitic lifestyle [5,11,20].

All tetrapyrrole pathway enzymes from the organisms mentioned above are encoded in the nucleus and hence must be targeted to a relevant compartment, after translation in the cytosol. For that purpose, cells evolved various targeting signals that can be N-terminal or C-terminal extensions, or lie internally within the protein [22]. For the transport through the ER, proteins are equipped with an N-terminal “signal peptide” (SP). Proteins targeted to plastids of primary phototrophs bear a “transit peptide” (TP) that is identified by translocons of outer and inner chloroplast membrane (TOC and TIC), respectively [23,24]. Complex plastids are coated with additional membranes; to pass them, proteins need a “bipartite targeting sequence” (BTS) consisting of a SP, that is cleaved immediately after crossing the outermost membrane, and a TP that escorts the protein to plastid stroma, where the TP is also excised to expose the mature protein [22,23,24,25,26].

*Chromera velia* is an alveolate alga, belonging to the group Apicomonada [27], isolated from stony corals from Sydney Harbor in Australia [28]. Together with *Vitrella brassicaformis*, it represents the closest known phototrophic relative to apicomplexan parasites [29]. Similar to other Apicomplexa and algae with complex plastids, both chromerids host rhodophyte-derived plastids surrounded by four membranes [28,29,30,31,32,33,34]. Although *C. velia* is a phototroph, it uses mitochondrially-located ALA synthase (ALAS) for the synthesis of ALA in the C4 route. All the C5 pathway enzymes found in other phototrophs are missing from chromerids [11]. The remaining enzymes of the pathway (from ALA to heme) display mosaic evolutionary origins (cyanobacterial, eukaryotic, and proteobacterial). Most of the enzymes involved in the pathway possess predicted bipartite targeting sequences (BTS) known to mediate import of nuclear-encoded proteins into complex plastids [11,35,36].

To see how the pathway is organized in the photosynthetic chromerids and to better understand what evolutionary forces shaped the unusual pathway in Apicomplexa, we experimentally tested the locations of heme pathway enzymes in the *C. velia*. As there is no transfection system for *C. velia* yet, we decided to use the heterologous expression in a photosynthetic diatom and in an apicomplexan parasite. This also allowed insight into the compatibility of targeting mechanisms between diatoms and apicomplexans, including chromerids. The best-established transfection systems in organisms related to *C. velia* are those for the apicomplexans *Toxoplasma gondii* and *Plasmodium falciparum*, and for the diatoms *Thalassiosira pseudonana* and *Phaeodactylum tricornutum* [37,38,39,40,41]. Both groups of organisms, apicomplexans, and diatoms contain secondary plastids surrounded by four membranes, and their plastid targeting mechanisms have been extensively studied [23,42,43,44,45,46,47,48]. The apicomplexan parasites are more closely related to *C. velia*; however, the plastids in *C. velia* were hypothesized to originate from a tertiary endosymbiotic event with a stramenopile [29,33,34,49,50]. Moreover, diatoms and *C. velia* share a phototrophic lifestyle, which requires more complex regulation of the tetrapyrrole synthesis due to the presence of the chlorophyll branch [5,15,16]. In this study, we localized six heme pathway enzymes from *C. velia* in the apicomplexan parasite *T. gondii* and in the diatom *P. tricornutum*: ALAS, two ALA dehydratases (ALAD1, ALAD2), uroporphyrinogen synthase (UROS) and two ferrochelatases (FECH1, FECH2). We also used specific antibodies generated against *C. velia* ALAS to localize this enzyme directly in *C. velia* cells by immunogold labeling and transmission electron microscopy.

## 2. Results

### 2.1. Prediction of Localization of Heme Synthesis Enzymes in C. velia

Various bioinformatics tools can be used to predict N-terminal targeting presequences typically associated with targeting to specific subcellular compartments. We analyzed the predicted targeting of the *C. velia* heme pathway enzymes using the following algorithms: SignalP 4.1 [51] in combination with TargetP 1.1 [52], to determine the presence of bipartite targeting sequences (BTS). As *C. velia* hosts complex plastid surrounded by four membranes [28,31], we also took advantage of the ASAFind predictor, designed to predict protein targeting to rhodophyte-derived complex plastids [53]. We ran ASAFind combined with different versions of SignalP and also used the C. velia optimized predictor ASAFind+ [54] in conjunction with SignalP 4.1. For mitochondrial transit peptides, we also used the prediction method MitoFates [55]. All results are summarized in the Appendix A.

According to SignalP 4.1 and TargetP 1.1, ALAS has no detectable ER signal peptide (ER-SP) or TP. This also applies to ALAD2 and UROS. Complete BTSs composed of SPs and TPs were found in ALAD1, porphobilinogen deaminase (PBGD), uroporphyrinogen decarboxylase 1 (UROD1), UROD2, both coproporphyrinogen oxidases (CPOX1, CPOX2), protoporphyrinogen oxidase 1 (PPOX1) and FECH1. ER-SPs without subsequent TP were found in UROD3 and FECH2. Mitochondrial TPs were detected in ALAD3 and PPOX2 by TargetP, while MitoFates predicted mitochondrial TPs for UROD1 and PPOX2 (all other enzymes were negative, results were identical regardless of the choice of organism group, Appendix A). Due to the good prediction performance of SignalP- and TargetP- based methods in diatoms [53,56] and *C. velia* [54], we decided to weight the results of SignalP/TargetP in conjunction with ASAFind or ASAFind+ higher than the MitoFates results.

All ASAFind predictions consistently suggested plastid localization for ALAD1, PBGD, UROD1, UROD2, UROD3, CPOX1, CPOX2, PPOX1, and FECH1. The remaining enzymes of the pathway appear to lack the ER-SP. The output of ASAFind and ASAFind+ combined with TargetP 2.0 agreed with the results mentioned above, except for FECH2, which according to TargetP 2.0, also has an ER-SP but no predicted plastid targeting by either ASAFind or ASAFind+. All above-mentioned predictors agreed on ALAS, ALAD2 and UROS lacking N-terminal targeting signal (Figure 2, Appendix A).

We interpret the results as follows: ALAS, ALAD2 and UROS have no detectable targeting signal. ALAD3 and PPOX2 have TP (detected by TargetP 1.1). The remaining enzymes (ALAD1, PBGD, UROD1, UROD2, UROD3, CPOX1, CPOX2, PPOX1, and FECH1) were predicted to be plastid-targeted proteins by most of the used predictors.

### 2.2. Analyses of C. velia Heme Pathway Enzymes N-termini Sequence

We analyzed the N-terminus sequence of *C. velia* heme pathway enzymes with predicted BTS. We compared the aa distribution and overall net charge of these proteins with works already published on the set of plastid targeted proteins from diatoms [53] and *C. velia* [54]. We found that *C. velia* has about 50% lower frequency of serine, and an overall higher proportion of positively charged residues within the first 20 aa of the TPs than diatoms (Figure 3). Seven of the nine predicted BTS of the *C. velia* enzymes of interest contain negatively charged residues that are almost absent in diatoms [53].

### 2.3. Localization of C. velia Heme Synthesis Enzymes via Heterologous Expression

For the heterologous reporter gene expression experiments, we selected six different genes from *C. velia*, which encode enzymes for four steps of the synthesis: ALAS synthesizes the first precursor of the pathway (ALA); ALAD catalyzes the condensation of two ALA molecules to the monopyrrole porphobilinogen; UROS represents the middle step of the pathway and forms the first macrocyclic tetrapyrrole—uroporphyrinogen III; FECH terminates the pathway by chelating the protoporphyrin IX with Fe^2+^ thus generating heme. Our attempts to heterologously express full-length *C. velia* genes showed toxicity for *P. tricornutum* (data not shown). Therefore, we used truncated genes to express only the N-terminal regions of the enzymes that included the targeting signals (if predicted), and some additional amino acids of the mature protein to end up with maximally 121 aa long sequence fused to an eYFP reporter gene (Figure 4).

#### 2.3.1. Localization in Phaeodactylum tricornutum

To express selected *C. velia* enzymes in *P. tricornutum*, we used two vectors, one bearing the gene of interest fused to the eYFP reporter and the other encoding the antibiotic resistance cassette, and co-transformed the diatom cells via micro-particle bombardment with a mixture of both vectors. Transformed genes are thought to be randomly integrated and stably maintained in the diatom genome [37]. After the antibiotic selection, we looked for eYFP positive cells using the fluorescence microscope and inspected them in detail via confocal microscopy. The signal from CvALAS- eYFP spanned through the diatom cell in the way typical for *P. tricornutum* mitochondria [57,58] and colocalized with the MitoTracker signal (Figure 5). The remaining enzymes, CvALAD1, CvALAD2, CvUROS, CvFECH1, and CvFECH2, consistently showed the so-called “blob-like” structures (Figure 5), a dense signal in close proximity to the plastid [35,59], with the same signal found even in the cases of CvALAD2, CvUROS, and CvFECH2, that lack SPs. The “blob-like” structure indicates targeting to the periplastidial space, between the two outermost and the two innermost membranes of the diatom complex plastid [60]. While the “blob-like” structure pattern was observed in the majority (88.9%) of cells in culture, 11.1% of cells showed co-localization of eYFP signals with ER-Tracker, indicating the presence of the enzyme in the ER (Appendix A). These results suggest that all tested enzymes, except for the mitochondria located CvALAS, are trapped either in the ER or in the periplastidial compartment of the diatom plastid, thus not entering the plastid stroma.

#### 2.3.2. Localization in *Toxoplasma gondii*

*Toxoplasma gondii* cells were transfected via electroporation with a vector bearing both, the chloramphenicol resistance cassette, and the *C. velia* heme pathway truncated gene, enabling fast selection of transfectants. In agreement with the *P. tricornutum* heterologous system, we also localized CvALAS in mitochondria of *T. gondii* (Figure 6) with the signal overlapping with the mitochondrial marker TgMys [47,61]. However, localization of CvALAD1, CvALAD2, and CvUROS in *T. gondii* conflicted with that found in *P. tricornutum.* Both CvALAD1 and CvALAD2 displayed cytosolic distribution in the apicomplexan model, while CvUROS was targeted to the mitochondrion. In agreement with the predictions, CvFECH1 localized to the apicoplast of *T. gondii*, but so did the CvFECH2, which has no predicted targeting signal (Figure 2).

### 2.4. Direct ALAS Localization via Immune Gold Labeling of C. velia Cells

We used a custom-made polyclonal rabbit antibody designed to detect *C. velia* ALAS (described in detail in materials and methods), to localize the enzyme on cell sections via immunogold labeling. As a control we used anti-βATPase [62]. Western blots on total protein extract from *C. velia* were performed prior to in vivo experiments to verify specificity of antibodies. The size of the mature ALAS protein of *C. velia* was estimated to be ~48 kDa (Protein Calculator v3.4; protcalc.sourceforge.net), our Western blot data showed a single band of approximately 42 kDa (Figure 7a). Anti-βATPase antibody was also tested on Western blot where we detected a signal of ~53 kDa (Figure 7d). Gold particles conjugated to secondary antibodies marking anti-CvALAS were in the majority (77%) of inspected *C. velia* sections detected in light-grey compartments of the cell (Figure 7b). Anti-βATP was in 61% detected in the same compartments as anti-CvALAS, beside it was also in 16% detected in plastids (Figure 7e). This finding was consistent in the majority of inspected *C. velia* sections (Figure 7c,f).

## 3. Discussion

To synthesize heme is crucial to the survival and growth of almost all living organisms. Two variants of heme biosynthesis pathways are known, the C4 pathway (in Alphaproteobacteria and most heterotrophic eukaryotes), and the C5 pathway (in Archaea, Eubacteria other than Alphaproteobacteria, and most phototrophic eukaryotes) [2]. Over the course of evolution, the specific localization of a particular enzyme is the result of multiple factors, including its evolutionary and endosymbiotic origin, which compartment has a major need of the resulting product and can also reflect pathway regulation, and/or the substrate availability [16,63,64]. *Chromera velia*, the closest known phototrophic relative to apicomplexan parasites, possesses a unique heme pathway, in which 5-aminolevulinic acid (ALA) is synthesized by the heterotrophic C4 pathway in mitochondria, like in apicomplexans parasites and primary heterotrophic eukaryotes. The downstream steps of the pathway were predicted to take place in the plastid [11]. We applied a combination of experimental and computational approaches to get a better insight into the heme biosynthesis in *C. velia* (Figure 8). Since a heterologous expression system of *C. velia* is not yet available, we decided to transfect more or less closely related well-established models, particularly the pennate diatom *P. tricornutum* [37,65,66] and the coccidian *T. gondii* [40,67] with the genes (or gene fragments) from *C. velia*.

Our results from both approaches (protein targeting predictions and heterologous transfections) were multivalent with a single exception for ALAS that constantly displayed mitochondrial localization. We also tested anti-CvALAS directly on *C. velia* section where the antibody was predominantly found in compartments that we assume to be mitochondria as the anti-βATP [62] localized to the same compartment. The anti-βATP that we used is regularly used as a mitochondrial marker in *Trypanosoma brucei* [62]. The ATP synthase is known to work in plastids of photosynthetic organism as well [68]; therefore, we detected a minor number of IG particles (16%) also in *C. velia* plastids (Figure 7). Mitochondrial localization of ALAS likely reflects the use of succinyl-CoA, the product of mitochondrial TCA cycle, as one of the substrates [11]. Although predictors failed to detect a mitochondrial TP in ALAS, the enzyme contains a presequence at the N-terminus when compared to the Alphaproteobacterial counterparts, showing some characters of mitochondrial TPs. Moreover, the pre-sequence contains two conserved heme-binding CP motifs that are shared with the ALAS sequences of animals and fungi, where the excess of heme blocks the ALAS import into the mitochondrion and thus inhibits the synthesis of ALA and heme [11]. ALA dehydratase (syn. porphobilinogen synthase) catalyzes condensation of two ALA molecules to constitute porphobilinogen [69]. Three ALAD pseudoparalogs were found in the *C. velia* genome [70] after the gradual refinements of gene models (CryptoDB database; http://cryptodb.org/cryptodb/app). Each pseudoparalog displays a different targeting pattern (Figure 2). The plastid localization of ALAD1 in *C. velia* was consistently suggested by all the predictors, reflecting its evolutionary origin in cyanobacteria [11]. ALAD2 seems to originate from the primary host nucleus (the nucleus of engulfed alga) [16]; however, we detected no targeting presequences here. The ALAD3, that was suspected to originate from the secondary host nucleus [16], lacks any ER-SP but contains a putative chloroplast TP. All predictors agreed in the plastid localization of PBGD while the following enzyme, UROS, has no detectable targeting signal. However, transfection in the diatom shows periplastid localization of UROS. UROS is always localized together with its accompanying enzymes: PBGD and UROD [5,15,16,64]. Such arrangement enables fast processing of hydroxymethylbiliane to uroporphyrinogen III. It was shown that if there is no UROS present during or immediately after the PBGD forms hydroxymethylbiliane, the biologically inactive isomer, uroporphyrinogen I (which is not a precursor of heme), forms spontaneously [71,72]. All three pseudoparalogs of UROD are putatively plastid targeted (Figure 2), despite their diverse evolutionary origin: UROD1 in cyanobacteria, UROD2 in the endosymbiont (primary host) nucleus, and UROD3 in the secondary host (exosymbiont) nucleus [11]. Although all three pseudoparalogs were predicted by ASAF and ASAFind+ prediction tools to be plastid targeted, SignalP 4.1 combined with TargetP 1.1 showed only low confidence for SP and no TP in UROD3 (Figure 2, Appendix A). Therefore, it is possible that at least two UROD (1, 2) enzymes are plastid located. There are two pseudoparalogs of CPOX found in *C. velia*. Again, all the predictions placed both CPOX to the plastid (Figure 2). PPOX and FECH were found to form a complex allowing efficient channeling of metabolites through the thylakoid membranes, which protects the highly reactive protoporphyrinogen IX [73,74]. Therefore, these enzymes should share the same compartment. Almost all the predictors coupled PPOX1 and FECH1 as plastid-targeted enzymes and PPOX2 with FECH2 as situated out of the plastid. SignalP 4.1+TargetP 1.1 suggests only PPOX1 as a plastid-targeted protein. The prediction algorithm did not find ER-SP in PPOX2, FECH2 was found to be SP positive; however, TP was not detected (Figure 2, Appendix A).

Transfections of *P. tricornutum* and *T. gondii* with selected *C. velia* heme pathway enzymes showed inconsistent results except for the mitochondrially located CvALAS. The mitochondrial TP was not found by any predictor in *C. velia* ALAS; however, the enzyme possesses N-terminal extension in its sequence [11] that directed the protein to mitochondria of both transfected organisms (Figure 5 and Figure 6). This finding suggests high versatility of the mitochondrial import machinery.

Transfections of *P. tricornutum* localized CvALAD1, CvALAD2, CvUROS, CvFECH1, and CvFECH2 outside the stroma of the diatom plastid, in the periplastidial space or the ER. These results may suggest that the diatom protein import machinery failed to recognize the TP domain in the *C. velia* enzyme or passed over the cleavage site between the ER-SP and the TP. The “blob-like” structure pattern of our constructs may, in some cases, phenotypically resemble peroxisome targeting [75]; however, peroxisome targeting in *P. tricornutum* relies on extreme C-terminal signal used by PTS1 import pathway, and the N-terminal depending import mechanism (PTS2) is not present at all [75]. In our work we used constructs formed predominantly by the N- terminus of *C. velia* heme pathway enzyme directly connected to eYFP (Figure 4). Therefore, we assume that the targeting into the peroxisome is not possible. The observed phenotype known as “blob-like” structure [35], has been explained as a block within the multistep plastid protein import pathway consisting of independent steps: Sec61 in cER, ERAD/SELMA in PPM and TOC and TIC in the outer and inner plastid envelope [26,76]. Plastid import in our constructs most probably stops before reaching the TOC complex of the second innermost membrane due to insufficient targeting signal within the sequence of *C. velia* heme enzyme constructs. This corresponds to the presence of the reporter protein in the periplastidial compartment (PPC), the space between the second and third plastid membranes. The reason for that could lie in the sequence of SP/TP motif of *C. velia* enzymes. SP leads the targeted protein through Sec61complex of the diatom chloroplast-ER membrane. Inside the lumen, the SP is cleaved off, and the TP is exposed to the translocon, which directs the protein across the second outermost membrane via the SELMA complex and through the TOC and TIC machinery of two innermost plastid membranes, respectively [26,76]. Kilian et al. [35] showed that possession of phenylalanine in position +1 of the TP is crucial for targeting the diatom plastid stroma. This specific phenylalanine requirement later broadened to required presence of F, Y, W and L [53,77,78]. Patron et al. [36] showed that this motif is well conserved among diatoms and brown algae. They also suggested that the ASA-F motif might be common for organisms with the rhodophyte-derived complex plastids. However, our analyses of *C. velia* heme pathway enzymes with predicted BTS have shown that the typical ASA-F motif is absent, and that negatively charged residues, which are almost absent in diatoms [53], are present in some *C. velia* heme pathway enzymes (Figure 3). It should be noted that the “blob-like” phenotype of the GFP accumulation in the diatom periplastidial compartment was first described as miss-targeting of plastid proteins with mutated BTS [35]. Later it was found that even a single amino acid substitution can change the targeting from the plastid to the periplastidial compartment [59,60]. This might explain the “blob-like” phenotype observed in our experiments because the BTSs of *C. velia* (Figure 3), in fact, resemble some of the mutated *P. tricornutum* BTSs, particularly in the case of changes in the TP net charges [78].

All investigated enzymes, except for ALAS, entered the periplastidial space of the diatom plastid or were captured in the ER just before crossing the second outermost plastid membrane. In other words, the transfected polypeptide successfully delivered eYFP over one (ER membrane) or two (periplastid membrane) outermost diatom plastid membranes but did not enter the plastid stroma. The SELMA translocon machinery, found in all rhodophyte-derived complex plastids with four membraned envelopes, is responsible for transporting protein across the second outermost membrane and mediates contact between the protein and TOC and TIC system of the two innermost plastid membranes [76]. We can speculate that if the proteins from *C. velia* contained a “proper” diatom ASA-F cleavage site, they would all end up in the stroma of the diatom plastid, in agreement with in silico predictions. On the other hand, ALAD2, UROS, and FECH2 show the same periplastid location in the diatom, even in the absence of a predictable BTS signal. As mentioned above, a minor fraction of the diatom transformants showed localization of proteins in the ER. As the ER is continuous with the outermost membrane of the diatom plastid, the observed pattern documents a failure to pass the second outermost membrane. That the enzymes reached the periplastidial space supports the presence of a functional ER-SP. Our results indicate the presence of a strict control mechanism controlling plastid protein import machinery of *P. tricornutum*.

In *T. gondii*, the cleavage site motif is less conserved than in algae [36]. Experimental localization of *C. velia* enzymes in *T. gondii* showed a more complex pattern (Figure 6). Both ALAD enzymes were located in the *T. gondii* cytosol with the signal of CvALAD2 displaying punctuated pattern distributed throughout the whole cell in a similar way as already described in [79,80]. The cytosolic localization suggests the inability of *T. gondii* translocon machinery to recognize the chromerid ER-SP, which is not surprising for CvALAD2 lacking a targeting presequence. On the other hand, even this enzyme was targeted to the diatom periplastidial compartment. However, without biochemical work that separates soluble and membrane fraction we cannot be 100% certain that both ALAD enzymes are localized only within *T. gondii* cytosol. The outermost membrane of the *T. gondii* apicoplast lacks, contrary to diatom plastid, a direct connection to ER [81]. Although the SP is also recognized by Sec61 translocon in the ER membrane, proteins are then transported via the ER and Golgi apparatus that directs protein-containing vesicles through the cytoplasm to the apicoplast [82]. After crossing the first membrane barrier, the remaining three membranes are equipped with a similar translocon system: periplastid membrane utilizes translocation ERAD/SELMA machinery [26,48,83,84,85,86], and both apicomplexans and diatoms employ homologous TOC and TIC plastid import machinery to transfer proteins over the outer and inner plastid membrane, respectively [48,87,88]. 

The mitochondrial targeting system seems to be more versatile. About 60% of mitochondrial proteins need to have positively charged amphipathic alpha-helical N-terminal presequence that is necessary for translocation through TOM and TIM mitochondrial membrane complexes. The remaining proteins do not carry cleavable presequence and rely on various internal targeting signal [89,90]. CvUROS was in *T. gondii* localized in mitochondria, contrary to its periplastidial localization in *P. tricornutum*. Despite the absence of any detectable targeting presequence *C. velia* UROS contains prolongation at the N-terminus when aligned with bacterial homologs (data not shown). Mitochondrial location of CvUROS in *T. gondii* demonstrates that the N-terminal presequence interpreted by the translocon machinery as mitochondrial TP is not always recognized by bioinformatic predictors. Both, *C. velia* FECHs were experimentally localized in the apicoplast of *T. gondii*. As mentioned above, the apicoplast is a minute organelle of approximately 0.15–1.5 μm in diameter [91]. Therefore, using confocal microscope, we are not able to distinguish whether the transfected CvFECHs arrived into the apicoplast stroma or remain trapped in any of the intermembrane spaces (similar to what we have seen in the diatom transfections) in immunofluorescence data. We hypothesize that *T. gondii* localizations of the *C. velia* enzymes are less likely to reflect real intracellular localizations in *C. velia,* due to the transport of intermediates over a high number of membranes (outlined in Figure 8).

The intracellular arrangement of the heme pathway in chromerids is non-canonical. Moreover, it seems to continue and terminate outside mitochondria [11]. Primary eukaryotic heterotrophs and some complex eukaryotic phototrophs are known to operate the tetrapyrrole pathway in different cell compartments. However, the first and terminal steps of the pathway usually locate in the same organelle, thus enabling easy pathway regulation [6,63]. There are two genes encoding terminal enzymes (FECHs) of the pathway in the genome of *C. velia*. One originates from a cyanobacterium, while the second is proteobacterial [11]. The corresponding proteobacterial FECH is in apicomplexan parasites mitochondrially targeted, while the cyanobacterial gene was lost during evolution. Multicellular plants also have two paralogue ferrochelatases originating in the gene duplication event. The first enzyme contains a C-terminal chlorophyll-binding domain and functions in the photosynthetic tissues. The second, which lacks the C-terminal domain, is utilized in the non-photosynthetic tissues such as roots [92,93]. However, the latter is also induced in photosynthetic tissues under various stress conditions [94,95]. We searched for a C-terminal chlorophyll-binding domain in both *C. velia* FECHs [96,97], but were not able to identify one. The reason for *C. velia* having two ferrochelatases in the plastid is therefore unknown. There are various mechanisms of tetrapyrrole synthesis regulation that work on different levels of the synthesis and together form a strong and sensitive network [95]. Among them, the heme-mediated feedback inhibition of ALA synthesis, which is conserved through different domains of life, plays a major role [10,61,98,99]. In various heterotrophs, ALAS, PPOX and FECH constitute the “heme metabolome complex”. The complex facilitates substrate channeling and coordinates tetrapyrrole metabolism [13,100]. However, the existence of a similar complex has not yet been proven in phototrophs; placement of these steps in different cellular compartments would require ambitious regulation and transport systems. Kořený et al. [11] found heme regulatory motifs in the sequence of *C. velia* ALAS, which indicates the presence of heme-mediated regulation of ALA synthesis. Therefore, we originally expected the location of proteobacterial FECH in the mitochondrion of *C. velia* (together with ALAS) as an intermediate state in the path to apicomplexan parasites, but the data do not support this hypothesis. All predictors agreed that FECH1 (the plastid originating pseudoparalog) possesses features typical for a plastid targeted protein, while the FECH2 should locate and outwith the plastid, but likely not in the mitochondrion. However, experimental localization showed both FECHs either in the diatom periplastidial compartment or the apicoplast. Therefore, we speculate that the possible role of the two ferrochelatases in a single cell could be protection of the cell under stress conditions.

While the tetrapyrrole pathway starts with the ALAS in the mitochondrion in chromerids, the remaining steps likely take place in the plastid. This model is further supported by the phylogenetic relationships among the individual enzymes of the pathway [11]. We summarized our findings in Table 1. The heterologous expression of *C. velia* ALAD1 and ALAD2 gave the same inconsistent results, placing the protein in the cytosol of *T. gondii* and PPC/ER in *P. tricornutum.* Despite that, we assume that ALAD1 is more likely a plastid-targeted protein, because our experimental results in *P. tricornutum* showed that the construct was transferred at least through the two outermost membranes of the diatom plastid. This, together with the combination of its cyanobacterial evolutionary origin, leads us to the conclusion that plastid localization is more plausible. The same cogitation was applied for FECH1 where the corresponding enzyme is also of cyanobacterial origin, and when heterologously expressed, it localized to PPC/ER of *P. tricornutum* and also to the apicoplast of *T. gondii*. We decided to conclude with an “uncertain localization” statement for ALAD3 and PPOX2 due to the absence of the experimental evidence and predictable ER signal peptides (see Appendix A for details), and their proteobacterial and eukaryotic origin, respectively. Both enzymes possess predicted mitochondrial transit peptides; however, particularly in PPOX, which makes a complex with FECH, its placement in the mitochondrion without FECH is unlikely. The localization of ALAD3 in the mitochondrion and a formation of porphobilinogen in this organelle would require additional transport of porphobilinogen to the plastid over its four membranes envelopes. The remaining enzymes (PBGD, UROD1, UROD2, UROD3, CPOX1, CPOX2, and PPOX1) were concluded as “plastid” localized due to the congruency of the prediction result. However, spatial separation of the beginning and the end of the pathway is unprecedented, and it would require regulatory mechanisms that are not yet known. Therefore, we cannot rule out the possibility of recent reassignments of intracellular locations or dual targeting of the enzymes. Our work on localization of *C. velia* heme pathway enzymes shows that the subcellular localization of biosynthetic pathway within any organism is a concert of multiple factors rather than a solo for one major element.

## 4. Conclusions

*C. velia* is a coral-associated alga bearing complex rhodophyte-derived plastid with a peculiar tetrapyrrole pathway. It synthesizes ALA using heterotrophic C4 path, however, which additionally supplies chlorophyll for photosystems. Using a combination of bioinformatics and experimental approaches we investigated the localizations of heme pathway enzymes in *C. velia*. Our data show that the pathway very likely starts in the mitochondrion with the remaining enzymes located to the plastid. We demonstrate that the proteins are targeted to various cellular compartments by stringent translocon mechanisms that are not universal even for evolutionary related organisms.

## 5. Materials and Methods

Targeting predictions and sequence analyses: Protein sequences of *C. velia* heme pathway enzymes as available at CryptoDB were used as input for all predictors used in this work. Prediction results of SignalP [101] and TargetP [102] were received from a web server (http://www.cbs.dtu.dk/services/), with SignalP 4.1 in “sensitive” mode. The ASAFind was used according to [53,103] and ASAFind+ was applied by modifying the original ASAFind code from Gruber et al. [53], according to the method described by Füssy et al. [54]. MitoFates [55] results were obtained from the MitoFates web service (http://mitf.cbrc.jp/MitoFates/cgi-bin/top.cgi). Sequence logos [103] and frequency plots were prepared using the WebLogo (http://weblogo.berkeley.edu/ [104]).

Cultivation conditions: *C. velia* (CCMP 2878) and *P. tricornutum* (CCMP 632) were grown in Guillard’s (f/2) medium (Sigma-Aldrich, St. Louis, MI, USA) in seawater and kept stationary in a 12/12 light/dark cycle regime at 26 °C and 18 °C, respectively [28,37]. Illumination during the light cycle was 100 μE m-2s-1. *Toxoplasma gondii* was grown in primary human foreskin fibroblasts (HFF) and treated as described in [46].

cDNA preparation and cloning: *C. velia* culture was harvested by centrifugation at 3000 rpm/10 min at 10 °C. The cell pellet was homogenized in TRI Reagent (Sigma-Aldrich, St. Louis, MI, USA) and total RNA was isolated following manufacturer’s instructions. cDNA was amplified from RNA with Superscript II reverse Transcriptase (Invitrogen, Thermo Fisher Scientific Inc, Waltham, MA, USA). Genes of the *C. velia* heme pathway were amplified from cDNA using specific primers designed for the Gateway cloning system (Invitrogen, Waltham, CA, USA). The amplified regions included the start codon and 201–360 bp downstream of the gene. Amplified genes were cloned into pENTR vectors (Invitrogen, Waltham, CA, USA) and verified by sequencing. pENTR vectors were subsequently recombined with pDEST-eYFP vectors by LR recombination reactions (Invitrogen, Waltham, CA, USA). Resulting pDEST-eYFP vectors contained the gene of interest followed by the sequence of the eYFP tag and were expressed under the fcp*b* promotor for *P. tricornutum* or under the tub promotor for *T. gondii* heterologous expression, respectively.

*Phaeodactylum tricornutum* heterologous expression: *P. tricornutum* cells were co-transfected with pDEST vector (fcp*b* promotor, *C. velia* heme pathway gene, eYFP tag) and pFCPFp-Sh ble vector (phleomycin resistance cassette). 1 μg/μL of vectors were mixed in a 1:1 ratio, mounted on tungsten (M-17) particles and introduced to the *P. tricornutum* nucleus by microparticle bombardment using the Biolistic PDS-1000/He Particle Delivery System (Bio-Rad, Hercules, Hercules, CA, USA). Cells were selected on 1% f/2 agar plates supplemented with 100 μg/mL phleomycin for 3–4 weeks in standard cultivation conditions. Phleomycin resistant colonies were subsequently transformed into liquid f/2 media and cultures of OD_600_ 0.2 were examined by fluorescent microscopy.

*Phaeodactylum tricornutum* fluorescent labelling of living cells: For mitochondria staining, 2 mL of *P. tricornutum* culture (OD_600_ 0.2) were incubated with 100 nM MitoTracker™ Orange CM-H2TMRos in standard cultivation conditions overnight. ER staining was done with ER-Tracker™ Red (BODIPY^®^ TR Glibenclamide) according to manufacturer’s instructions. A total of 0.1 μg/mL DAPI was used to incubate with cells for 15 min in dark. All chemicals used for staining were from: Thermo Fisher Scientific Inc, Waltham, MA, USA. Prior to confocal microscopy, 1ml of cells were harvested by centrifugation (6000 rcf/10 min/room temperature), washed, and resuspended in 100 µL of PHEM buffer.

*Toxoplasma gondii* transient transfection: *T. gondii* RH strain tachyzoites were purified from suspension using 3-μm-pore size polycarbonate filters, spun down by centrifugation (15,000 rcf/20 min/11 °C) and resuspended in electroporation buffer [105,106,107]. A total of 300 μL of parasites (appx. 10^7^) and 20 μL of plasmid DNA (4.5 ng/μL) were transferred to a sterile electroporation cuvette and electroporated (1500 V, 25 Ω). The whole volume of the cuvette was poured into a well containing coverslips with confluent monolayer of HFF cells. Parasites were fixed and examined by immunofluorescence assay after 3 days of cultivation.

*Toxoplasma gondii* immunofluorescence assay: HFF-covered 12 mm round coverslips were inoculated with transfected *T. gondii* and grown for 24–72 h in standard growing conditions. Growing media was replaced with 4% paraformaldehyde in PBS and incubated for 20 min to fix parasites at RT. Permeabilization was done in 0.25% Triton X-100 in PBS for 20 min at RT. Coverslips were blocked in 1% BSA in PBS for 20 min before incubation in primary antibody (anti-TgMys 1:1000, anti-GFP 1:200, anti ROM 4 1:1000; all in 1% BSA/1xPBS) for 1 h at RT. Coverslips were washed in 1% BSA in Triton X-100 in PBS three times and incubated with secondary antibody for 30 min at RT. A final wash in 1% BSA in Triton X-100 in PBS three times was conducted, and coverslips were mounted.

Microscopy: Cellular localizations were analyzed in both transfection systems and *C. velia* with the Fluo View^TM^ 1000 confocal system configured with an inverted mobile IX81 microscope (Olympus, Tokyo, Japan). A scanning laser with wavelength 515 nm was used for excitation of chlorophyll and eYFP. The emission spectra were detected using the following bandwidths: DAPI 345–455 nm, eYFP 525–571 nm, chlorophyll 620–710 nm, and MitoTracker^®^ Orange CM-H_2_TMRos 554–576 nm, ER-Tracker™ Red (BODIPY^®^ TR Glibenclamide) 590–640 nm. All chemicals used for staining were from: Thermo Fisher Scientific Inc., Waltham, MA, USA. Images were processed using Olympus FV10-ASW software and Imaris (Olympus, Tokyo, Japan).

Oligopeptide selection for antibodies production: Antibodies for direct localization *C. velia* ALAS were generated by Clonestar s.r.o. (Brno, Czech Republic) using synthetic oligopeptide conjugated to KHL/BSA. The oligopeptide sequence was chosen according to sequence analysis with Dnastar Lasergene Protean software suite version 7.1 (Madison, WI, USA), followed by analysis of conserved motives in Geneious (Biomatters Ltd., Auckland, New Zealand). Candidate oligopeptides were mapped to known tertiary structures on the NCBI server. Oligopeptide sequence (14 aa) with the most plausible epitope and surface probability, conservation in sequences alignment and surface mapping on tertiary structure was chosen.

SDS-PAGE and Western blotting: For SDS-PAGE and Western blotting we used the Bio-Rad Mini-Protean tetra cell system according to manufacturer’s instructions. A total of 2–8 μL of *C. velia* protein lysate was loaded on 5/12% SDS-PAGE gel and then transferred to a PVDF membrane (GE Healthcare Life Sciences, Chicago, IL, USA). The membrane was then blocked with 5% nonfat dry milk in TBS and incubated for 1 h with the primary antibody (1:5000) in blocking solution containing 0.2% Tween 20 (TTBS). The membrane was washed three times with TTBS and incubated for 1 h with Anti-Rabbit Immunoglobulins/HRP (Dako, Glostrup, Denmark) (dilution 1:1300). Chemiluminescence reactions were performed using Clarity Western ECL Substrate (Bio-Rad). The expected size of enzyme was determined based on the protein sequence using online software Protein Calculator v3.4 (protcalc.sourceforge.net).

Immuno-gold labelling and Transmission electron microscopy: For antibody labelling, samples were blocked by placing the nickel grids with ultra-thin sections of *C. velia* on a drop (30 µL) of blocking/wash buffer (3% BSA, 0.1 M HEPES pH 7.4, 0.05% Tween-20) for one hour. The grids were moved to a drop of blocking/wash buffer containing Rabbit IgG anti-ALAS antibody (1:40), or anti-βATP (1:40) for 15 min and washed with a drop of blocking/wash buffer six times for 15 min each. Secondary immunolabelling was done with protein A conjugated to 15 nm gold, diluted 1:50 in blocking/wash buffer, for one hour. Labelling was followed by six washing steps with a drop of blocking/wash buffer, each 15 min, and finally grids were rinsed two times rinse with a drop of deionized H_2_O and dried on paper. Post-contrasting was done in a drop of saturated Uranyl-Acetate/ethanol for 12 min. The grids were washed with 30% ethanol 3 times each for 90 s and finally dried on paper. All preceding steps were completed at room temperature. Images were obtained with a transmission electron microscope (JEM 1010, JEOL Ltd., Tokyo, Japan) at an acceleration voltage of 80 kV.

Quantification of immune-gold labeling distribution: Immuno-gold (IG) labeling of *C. velia* was quantified according to the method described in [108,109]. IG particles from a set of 35 micrographs of both IG labeling (anti-CvALAS, anti-βATP) of the same magnitude (40,000×) was quantified using the ImageJ software (https://imagej.nih.gov/) using a grid with cross distance 280 nm. Number of IG particles was counted (for each compartment) as follows:IG number = ∑ gold particles(1)

Total area of each compartment: mitochondria, plastid, nucleus and other (=remaining organelles, vacuoles and cytoplasm) was estimated as follows:area nm^2^ = ∑ P × d × d(2)
where “P” means points (crosses) hits and “d” means the distance between crosses in the grid used in ImageJ software.

## Figures and Tables

**Figure 1 ijms-22-06495-f001:**
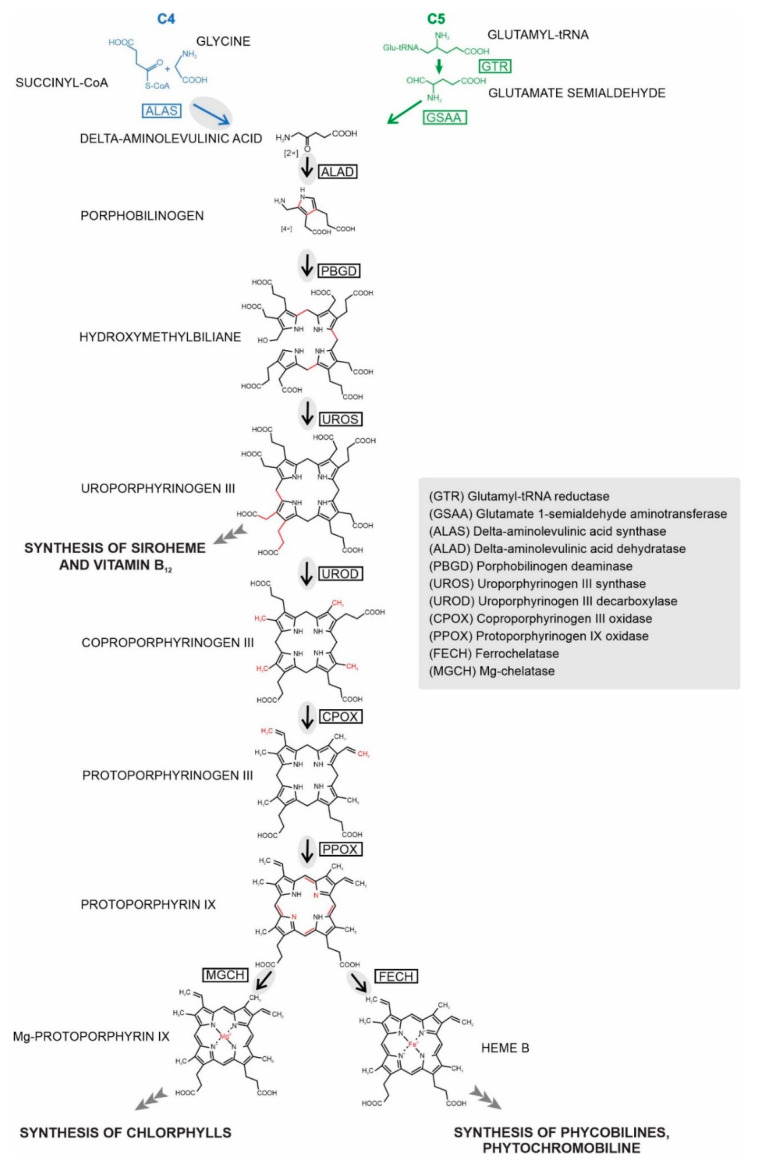
Tetrapyrrole synthesis. Enzymes working in particular steps of the synthesis are denoted by acronyms in boxes with their full names explained in the grey panel. Enzymatic steps (arrows) present in *C. velia* are in the grey oval. Changes in product structure are highlighted in red.

**Figure 2 ijms-22-06495-f002:**
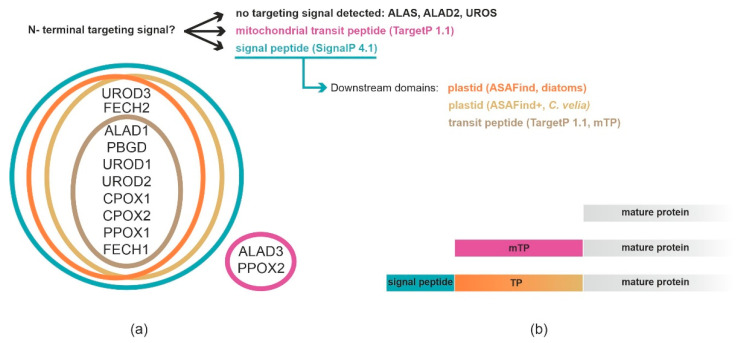
In silico targeting predictions for heme biosynthesis pathway enzymes in *C. velia*. (**a**) Euler diagram displays interpretation of targeting signals by various predictors. (**b**) Scheme showing different possibilities of N-terminal targeting signals.

**Figure 3 ijms-22-06495-f003:**
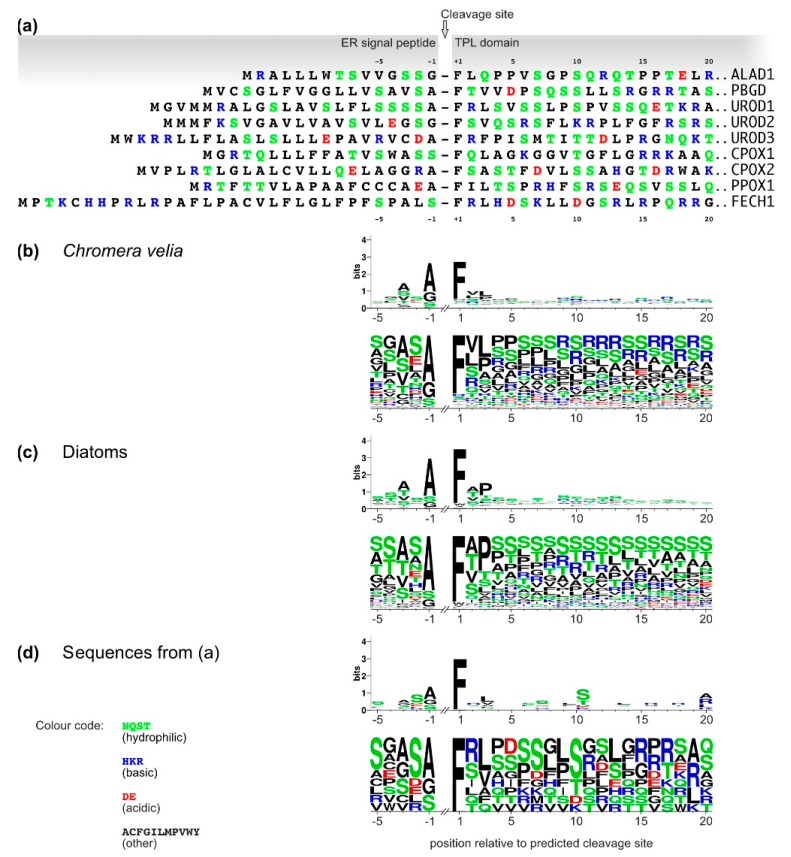
(**a**) ER-SP and TP domains of *C. velia* heme pathway enzymes. Coordinates are relative to the predicted SP cleavage site (arrow). Only enzymes that were positive for BTS are shown, amino acids in one letter code, color code is identical for all panels. (**b**–**d**) Sequence logos (upper panels) and frequency plots (lower panels) of plastid targeting BTS cleavage site motifs and TPs from (**b**) *C. velia* (*n* = 146 data from [5]), (**c**) diatoms (*n* = 166, reproduced from [53]), and (**d**) the *C. velia* heme pathway enzymes shown in A (*n* = 9).

**Figure 4 ijms-22-06495-f004:**
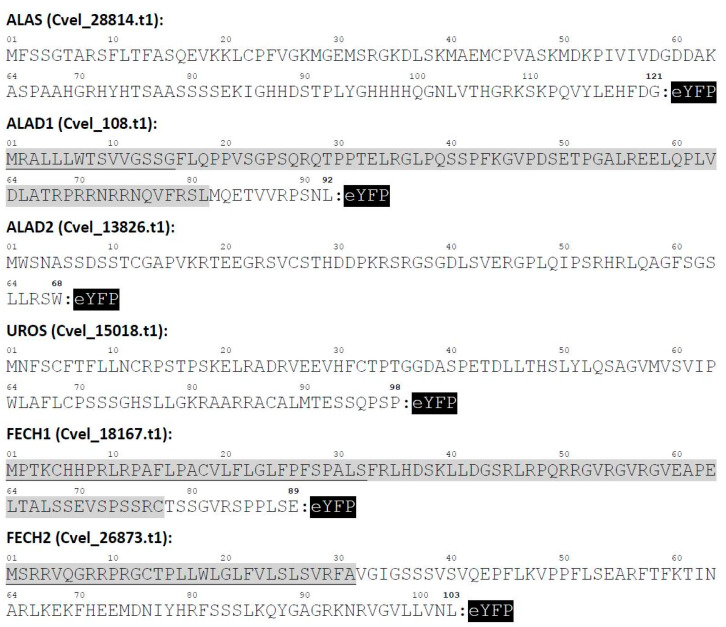
Constructs for heterologous expression of *P. tricornutum* and *T. gondii* with truncated *C. velia* proteins. Numbers above the aa sequence correspond to amino acid position in the protein. Presequences are marked in grey, predicted SPs are underlined. Enhanced yellow fluorescent protein used to tag the construct is displayed as acronym (eYFP) in a black box. CryptoDB accession numbers of proteins are given in parenthesis behind the protein name.

**Figure 5 ijms-22-06495-f005:**
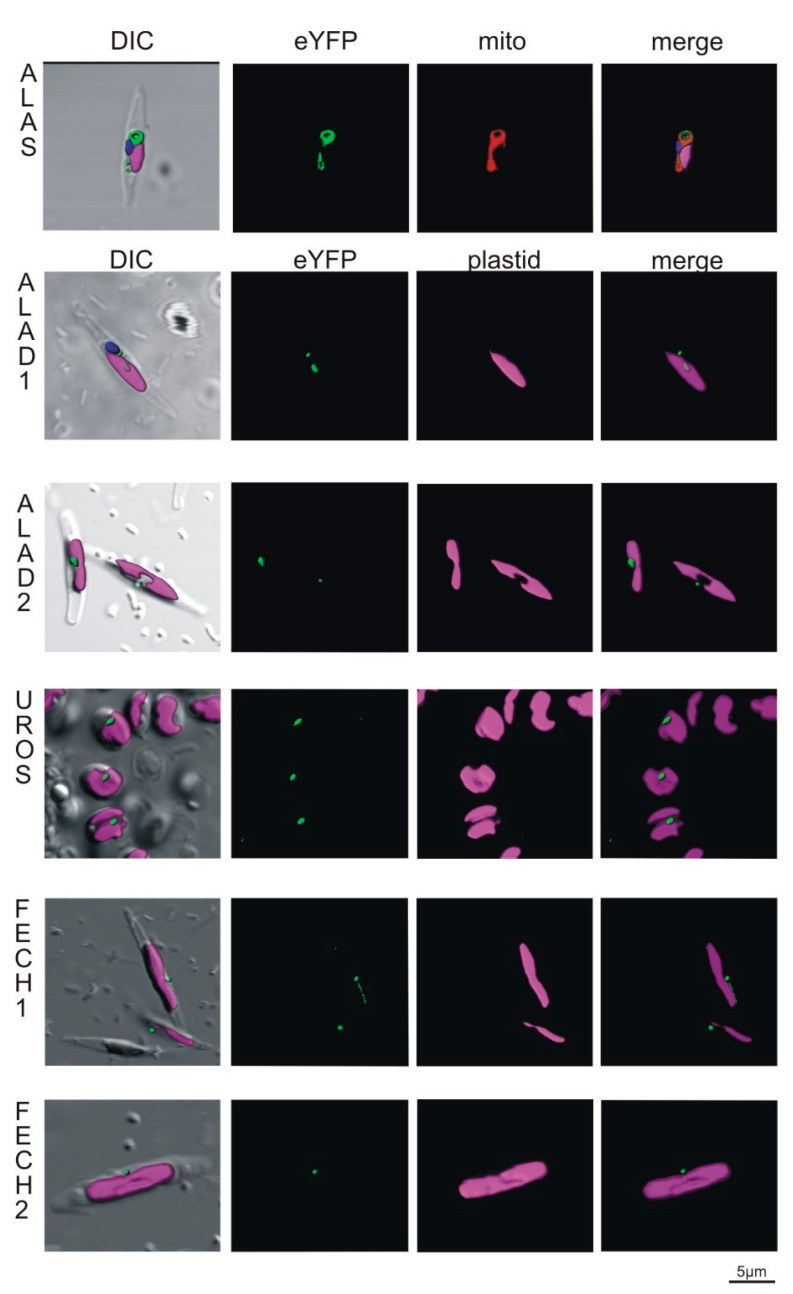
Heterologous expression of *Phaeodactylum tricornutum* with genes from *Chromera velia* heme pathway enzymes. Selected enzymes were tagged on their C-terminus by eYFP (green), magenta indicates plastid autofluorescence, MitoTracker^®^ Orange CM-H2TMRos (ALAS, red) indicates mitochondrion. Green eYFP signal of *C. velia* ALA synthase colocalizes with red signal of *P. tricornutum* mitochondrion (row ALAS). Typical “blob-like” structures are found in heterologous expression of ALA dehydratases (ALAD1, ALAD2), uroporphyrinogen synthase (UROS) and both ferrochelatases (FECH1, FECH2).

**Figure 6 ijms-22-06495-f006:**
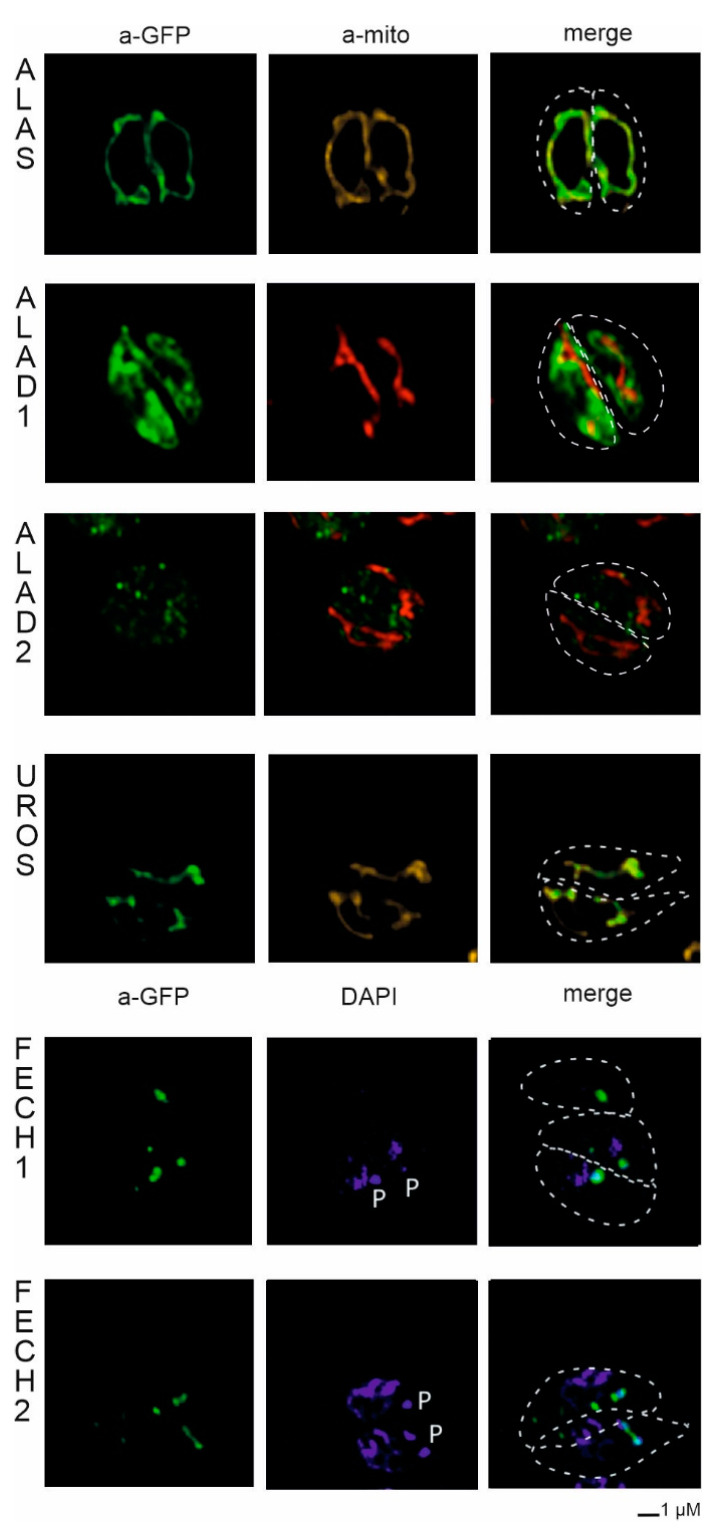
Heterologous expression of *Toxoplasma gondii* with genes from *Chromera velia* heme pathway enzymes. Immunofluorescence assays of transfected *T. gondii*, anti-GFP antibody were used to detect eYFP tagged *C. velia* enzymes. Anti-GFP (green) colocalized with mitochondrial anti-TgMys (a-mito, red and yellow) signal in case of ALAS and UROS. ALAD1 and ALAD2 signal were detected in the cytosol. FECH1 and FECH2 signal was found to overlap with DAPI (blue) signal at the area of parasite apicoplast. Apicoplast is denoted by “P”. Dashed line indicates *T. gondii* cell border.

**Figure 7 ijms-22-06495-f007:**
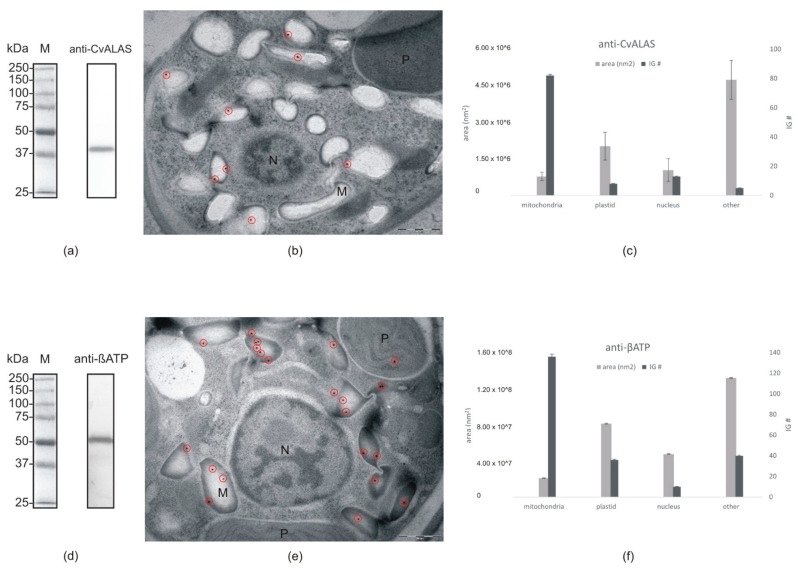
Immunogold labeling: (**a**) Western blot with anti-CvALAS on total protein extract from *C. velia* (**b**) Micrograph of *C. velia* ultrathin section after immunogold labeling with specific anti-CvALAS as a primary antibody. The majority of gold particles (encircled) were detected in the mitochondria. (**c**) Distribution of secondary IG particles (detecting anti-CvALAS) among cell compartments counted from all 35 micrographs. (**d**) Western blot with anti-βATP on total protein extract from *C. velia* (**e**) Micrograph of *C. velia* ultrathin section after immunogold labeling with specific anti-βATP as a primary antibody. The majority of gold particles (encircled) were detected in *C. velia* mitochondria. (**f**) Distribution of secondary IG particles (detecting anti-βATP) among cell compartments counted from all 35 micrographs. N = nucleus, M = mitochondria, P = plastid.

**Figure 8 ijms-22-06495-f008:**
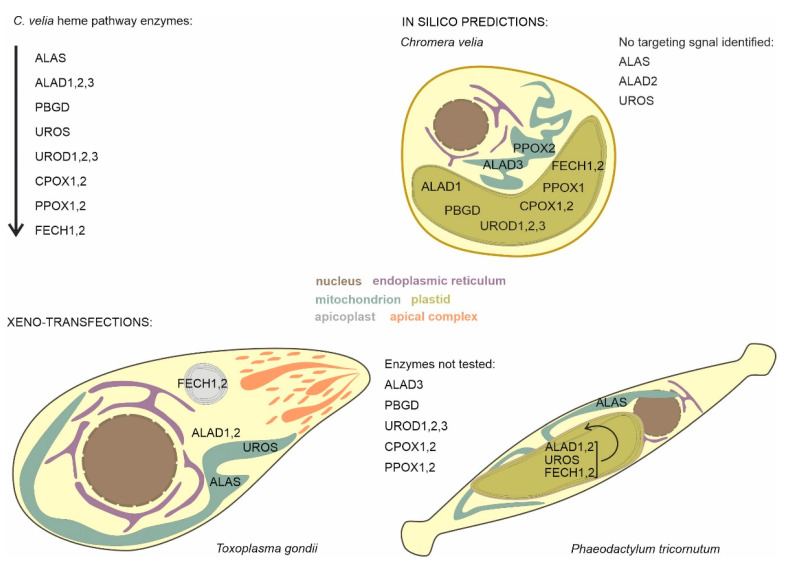
Intracellular distribution of heme biosynthesis in *Chromera velia*, inferred from predictions, and heterologous reporter gene expression in *Toxoplasma gondii* and *Phaeodactylum tricornutum*.

**Table 1 ijms-22-06495-t001:** Table showing results described in this manuscript. Enzymes are listed according to their order during the synthesis of heme. Evolutionary origins of each of the enzymes are based on phylogenetic analyses from the work of [11,16]. The last column of the table contains our hypothetical conclusions about the *C. velia* enzyme localization based on our findings.

Enzyme	Accession (CryptoDB)	Evolutionary Origin	Targeting Prediction	Localization *T. gondii*	Localization *P. tricornutum*	Conclusion
**ALAS**	Cvel_28814.t1	Alphaproteobacteria	No targeting signal identified	Mitochondria	Mitochondria	Mitochondria
**ALAD1**	Cvel_108.t1	Cyanobacteria	Plastid	Cytosol	PPC/ER	Plastid
**ALAD2**	Cvel_13826.t1	Primary alga	No targeting signal identified	Cytosol	PPC/ER	Uncertain location
**ALAD3**	Cvel_36189.t1	Proteobacterial	Mitochondria	Not tested	Not tested	Uncertain location
**PBGD**	Cvel_26028.t1	Alphaproteobacteria	Plastid	Not tested	Not tested	Plastid
**UROS**	Cvel_15018.t1	Uncertain origin in primary alga	No targeting signal identified	Mitochondria	PPC/ER	Uncertain location
**UROD1**	Cvel_14720.t1	Cyanobacteria	Plastid	Not tested	Not tested	Plastid
**UROD2**	Cvel_5098.t1	Endosymbiont nucleus	Plastid	Not tested	Not tested	Plastid
**UROD3**	Cvel_31936.t1	Secondary host nucleus	Plastid	Not tested	Not tested	Plastid
**CPOX1**	Cvel_21486.t1	Secondary host nucleus	Plastid	Not tested	Not tested	Plastid
**CPOX2**	Cvel_2641.t1	Uncertain origin in primary alga	Plastid	Not tested	Not tested	Plastid
**PPOX1**	Cvel_13840.t1	Cyanobacteria	Plastid	Not tested	Not tested	Plastid
**PPOX2**	Cvel_18037.t1	Eukaryotic origin	Mitochondria	Not tested	Not tested	Uncertain location
**FECH1**	Cvel_18167.t1	Cyanobacteria	Plastid	Apicoplast	PPC/ER	Plastid
**FECH2**	Cvel_26873.t1	Alphaproteobacteria	Signal peptide positive	Apicoplast	PPC/ER	Uncertain location

## Data Availability

Data available in a publicly accessible repository that does not issue DOIs. Publicly available datasets were analyzed in this study. These data, all sequences used in this work, are available at: CryptoDB (Cryptosporidium Informatics Resources) at https://cryptodb.org/cryptodb/app/, according to their accession number.

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
