# Peer review of "Using Diatom and Apicomplexan Models to Study the Heme Pathway of Chromera velia"

_ijms, 2021, doi:10.3390/ijms22126495_

Round 1

Reviewer 1 Report

This paper written by Richtova and colleagues demonstrates the complex pattern of localization of the enzymes for the heme synthesis pathway in the photosynthetic close relative of apicomplexan parasites, Chromera veria. The authors expressed the recombinant proteins of GFP N-terminally tagged by the N-terminal regions of some of the Chromera heme enzymes in the model diatom Phaeodactylum and the model apicomplexan Toxoplasma. They found that the ALAS of Chromera, the first step enzyme of heme pathway, localized in mitochondria in both systems, although it lacks a detectable mitochondria-targeting transit peptide at the N-terminus. This finding was confirmed by the immuno-TEM assay with Chromera cells demonstrating that the gold particle-labeled ALAS antibody localized in the mitochondria of Chromera.

In contrast, other enzymes showed controversial localization distinct between the two expression systems, which were also different from the in silico prediction performed previously by the same authors. This work provides insight into evolution and diversity of localization and regulation of the Heme synthesis pathways in eukaryotes and also how difficult it is to interpret any localization studies with heterologous expression systems if inconsistent with in silico prediction.

The manuscript is well written and provides insightful data worth being published. However, some parts of discussion and interpretation of results seem to need more detailed explanation and discussion, which would help the manuscript more readable.

Major points

Interpretation of Fig. 3

Some of the eYFP signals resemble those of peroxisomes as in previous studies (e.g., Gonzalez et al. 2011. PLoS ONE 6(9): e25316). Especially, signals for ALAD1 and FECH1 seem to be localized outside plastids, like peroxisomes. I understand that the recombinant proteins unlikely possess potential to be localized in persoxisomes, given the C-terminal sequence of eYFP which does not show the diatom peroxisome targeting signal. But I recommend rationalizing more explicitly why those signals can be interpreted as those for the PPC localization of diatom proteins.

Interpretation of Fig. 4

Although I agree with the cytosolic localization of ALAD1 in the Toxoplasma system, I am not sure whether ALAD2 actually localizes the cytosol. The eYFP signal seems to show localization in certain organelles. As I am not familiar with the Toxoplasma system, the authors are recommended explaining why this demonstrates the cytosolic localization.

                 In addition, for FECH1 and FECH2, I am not sure why colocalization with DAPI signals are evidence of the apicoplast localization. It is required for the authors to explain how they regard the DAPI signals are of apicoplasts. Is it a canonical way for Toxoplasma cells? Relevant to this figure, as I cannot see the “apicoplast” DAPI signals when I print it out, although there are the labels “P” for apicoplasts, a clearer picture is required.

Discussion in P11-12 and Fig. 7

The analyses of N-termini for the heme pathway enzymes should be mentioned in the Results section and the overall structure of the Discussion section should be reformatted according to that.

Conclusion in Table 1

I think that the conclusion for the localization in Table 1 is not rationalized and more detailed discussion is required. For ALAD2, UROD1, and FECH2, the conclusion for localization “uncertain” seems reasonable given inconsistent results from two heterologous expression systems and in silico prediction. Nevertheless, the authors conclude the plastid localization of ALAD1 and FECH even with inconsistent results of the localization studies and in silico prediction. Similarly, ALAD3 and PPOX2 would be concluded as mitochondrial if relying only on the in silico prediction as for PPGD, UROD1, UROD2, UROD3, CPOX1, CPOX2, and PPOX1. More detailed discussion and explanation of why the authors have concluded as shown in Table 1.

Minor points

line 19

I think Chromera is not an apicomplexan.

the N-terminus of ASAS

Other prediction programs may detect the mitochondrial targeting signal, such as MitoFates and NommPred. They might be worth trying.

line 512

Is this title proper for the experiment? I cannot find any procedure with an antibody in the observation of P. tricornutum cells.

Reviewer 2 Report

The manuscript is well written and well conceived. The topic  and the in depth analysis performed by the authors is commendable. Some suggestions from my end would be:

  1. Mind the technical jargons, there are several word choice errors in teh4 manuscript. please have a second look at them. Use established terms instead.
  2. Line # 27- Downstream enzymes?
  3. In Line 17 you please rephrase to reflect that you are talking about the cellular location of the enzymes. 
  4. I would advise the authors to make a graphical representation of the pathway mentioned in the introduction for ease.

Author Response

Response to Reviewer 2 Comments

  1. Mind the technical jargons, there are several word choice errors in teh4 manuscript. please have a second look at them. Use established terms instead.

Thank you for pointing this out, we carefully revised the manuscript. During the revisions, we eliminated unnecessary jargon and also improved the English where needed. We hope that the text is now in an acceptable form.

  1. Line # 27- Downstream enzymes?

Replaced with “In P. tricornutum all remaining enzymes, from ALA-dehydratase to ferrochelatase, were placed either in the endoplasmic reticulum or in the periplastidial space”

  1. In Line 17 you please rephrase to reflect that you are talking about the cellular location of the enzymes.

This sentence now reads: "Instead, the final subcellular location of the enzyme reflects multiple factors, including evolutionary origin, demand for the product, availability of the substrate, and mechanism of pathway regulation."

  1. I would advise the authors to make a graphical representation of the pathway mentioned in the introduction for ease.

Thank you for this suggestion, we added Figure 1 which also contains the chemical structures of the compounds.